# Early Life Antimicrobial Exposure: Impact on *Clostridioides difficile* Colonization in Infants

**DOI:** 10.3390/antibiotics11070981

**Published:** 2022-07-21

**Authors:** Chinwe Vivien Obiakor, Jaclyn Parks, Tim K. Takaro, Hein M. Tun, Nadia Morales-Lizcano, Meghan B. Azad, Piushkumar J. Mandhane, Theo J. Moraes, Elinor Simons, Stuart E. Turvey, Padmaja Subbarao, James A. Scott, Anita L. Kozyrskyj

**Affiliations:** 1School of Public Health, University of Alberta, Edmonton, AB T6G 1C9, Canada; obiakor@ualberta.ca; 2Faculty of Health Sciences, Simon Fraser University, Burnaby, BC V5A 1S6, Canada; jaclyn_parks@sfu.ca (J.P.); ttakaro@sfu.ca (T.K.T.); 3Cancer Control Research, BC Cancer Research Institute, Vancouver, BC V5Z 1L3, Canada; 4School of Public Health, University of Hong Kong, Hong Kong; heinmintun@ualberta.ca; 5The Jockey Club School of Public Health and Primary Care, The Chinese University of Hong Kong, Hong Kong; 6Microbiota I-Center (MagIC), The Chinese University of Hong Kong, Hong Kong; 7Dalla Lana School of Public Health, University of Toronto, Toronto, ON M5T 3M7, Canada; nadia.morales@utoronto.ca (N.M.-L.); james.scott@utoronto.ca (J.A.S.); 8Department of Pediatrics & Child Health, University of Manitoba, Winnipeg, MB R3A 1S1, Canada; meghan.azad@umanitoba.ca (M.B.A.); elinor.simons@umanitoba.ca (E.S.); 9Department of Pediatrics, University of Alberta, Edmonton, AB T6G 1C9, Canada; mandhane@ualberta.ca; 10Department of Pediatrics, University of Toronto, Toronto, ON M5G 1X8, Canada; theo.moraes@sickkids.ca (T.J.M.); padmaja.subbarao@sickkids.ca (P.S.); 11Department of Pediatrics, University of British Columbia, Vancouver, BC V6H 0B3, Canada; sturvey@cw.bc.ca

**Keywords:** antibiotics, cleaning products, antimicrobials, *Clostridioides difficile*, *C. difficile*, infant, gut microbiota

## Abstract

The relationship between antibiotic use and *Clostridioides difficile* (*C. difficile*) has been well established in adults and older children but remains unclear and is yet to be fully examined in infant populations. This study aimed to determine the separate and cumulative impact from antibiotics and household cleaning products on *C. difficile* colonization in infants. This study included 1429 infants at 3–4 months of age and 1728 infants at 12 months of age from the Canadian Healthy Infant Longitudinal Development (CHILD) birth cohort. The levels of infant antimicrobial exposure were obtained from hospital birth charts and standardized questionnaires. Infant gut microbiota was characterized by Illumina 16S ribosomal ribonucleic acid (rRNA) gene sequencing. Analysis of *C. difficile* was performed using a quantitative polymerase chain reaction (qPCR). Overall, *C. difficile* colonized 31% and 46% of infants at 3–4 months and 12 months, respectively. At 3–4 months, *C. difficile* colonization was significantly higher in infants exposed to both antibiotics and higher (above average) usage of household cleaning products (adjusted odds ratio (aOR) 1.50, 95% CI 1.03–2.17; *p* = 0.032) than in infants who had the least antimicrobial exposure. This higher colonization persisted up to 12 months of age. Our study suggests that cumulative exposure to systemic antibiotics and higher usage of household cleaning products facilitates *C. difficile* colonization in infants. Further research is needed to understand the future health impacts.

## 1. Introduction

Global antibiotic consumption has risen substantially over the past two decades [1]. While the administration of antibiotics directly to infants has reduced by more than 50% in Canada [2], an overlooked source of infant antibiotic exposure is the administration of antibiotics to mothers during childbirth. The guidelines of the Society of Obstetricians and Gynecologists of Canada (SOGC) recommend antibiotic prophylaxis before caesarean section or during labor (intrapartum antibiotic prophylaxis, IAP) for women who are positive for Group B *Streptococcus* (GBS) or who have other risk factors [3]. In accordance with these recommendations, up to 40% of newborns are exposed before or during delivery to maternal IAP [4]. Furthermore, about 2% to 5% of vaginally delivered newborns receive intravenous (IV) antibiotics after birth for treatment of suspected neonatal sepsis [4,5]. Outside of the hospital setting, household standards of cleaning have also evolved over the years, in response to various socio-cultural factors. The commercialization of the cleaning industry has encouraged the increased use of cleaning products in the home, contributing to antimicrobial exposure [6].

Antimicrobial exposure during infancy is not without consequences. Epidemiological studies have shown that early life antimicrobial exposure is associated with disruption of gut microbiota [7,8,9,10] and influences future asthma and allergic diseases [11,12]. Multiple courses of antibiotics have a great influence on the composition of infant gut microbiota [13]; however, cumulative antimicrobial exposure from additional sources, such as household cleaning products, have not been studied. *Clostridiodes* (formerly *Clostridium*) *difficile* (*C. difficile*) is a gram-positive spore forming bacteria. It is a major pathogen that is responsible for the clinical manifestations of antibiotic-induced diarrhea in adults and older children [14]. Although the colonization rate is high (over 40%) in infants below the age of 1 year, the biological relevance of *C. difficile* in this age group remains uncertain, as most colonized infants do not manifest clinical symptoms [15]. However, colonization in infancy may serve as a reservoir for adult *C. difficile* infections (CDI) or a marker for reduced colonization resistance and delayed gut microbiota maturation [16,17,18]. Disruption of the gut microbiome early in life may be associated with conditions such as allergy and asthma, inflammatory bowel disease (IBD), and obesity later in life [19]. Antimicrobial exposure can destroy the diversity of the gut microbiome, limiting the number of microbes that are in competition for growth, thereby allowing *C. difficile* colonization and overgrowth. Previous studies proposed clear effects of antimicrobial exposure on the infant gut microbiota, but those studies were limited to small samples or reporting at the genus level [7,8,9,10]. The aim of our study was to determine the separate and cumulative impact of antibiotics and environmental antimicrobials (i.e., household cleaning products) on *C. difficile* colonization, and to understand how these antimicrobial exposures modify the infant gut microbiota. This area of research is currently understudied, and the issues are not fully understood.

## 2. Results

### 2.1. Study Population

In study infants at 3 months of age, 29% of them were exposed to no antibiotics and lower (below average) usage of cleaning products (**NALC**); 24% were exposed to any antibiotics and lower (below average) usage of cleaning products (**AALC**); 22% were exposed to no antibiotics and higher (above average) usage of cleaning products (**NAHC);** and 25% were exposed to any antibiotics and higher (above average) usage of cleaning products (**AAHC**) (Table 1). All caesarean section (CS) deliveries by participants in the Canadian Healthy Infant Longitudinal Development (CHILD) study received antibiotic prophylaxis, in accordance with Canadian practice guidelines. Fecal samples were collected from 1429 infants at 3–4 months of age (mean age 3.6 ± 1.04 months) and 1728 infants at 12 months of age (mean age 12.2 ± 1.48 months). Of note, the 12-month sample was larger because stool samples from participants were easier to collect for analyses. In general, our sample of infants at both 3–4 months of age and 12 months of age did not differ from the overall CHILD cohort (Appendix A).

### 2.2. Study Population and C. difficile Colonization

#### 2.2.1. *C. difficile* Colonization at 3–4 Months of Age

In our study, the prevalence of *C. difficile* colonization in infants at 3–4 months of age was 31% (445/1429). The *C. difficile* colonization of infants differed according to antimicrobial exposure: 24% for NALC, 30% for AALC, 32% for NAHC, and 39% for AAHC (*p* < 0.001; Figure 1). Before adjusting for covariates, the odds of *C. difficile* colonization were 38% higher (odds ratio (OR): 1.38, 95% confidence interval (CI) 1.00–1.91; *p* = 0.047) in the AALC infants, 52% higher (OR: 1.52, 95% CI 1.10–2.11; *p* = 0.011) in the NAHC infants, and 103% higher (OR: 2.03, 95% CI 1.49–2.78; *p* < 0.001) in the AAHC infants, compared with the NALC infants (Table 2). The initial selection of covariates for model testing is shown in the directed acyclic graph (DAG; Appendix A). After adjusting for covariates in the final model (i.e., maternal age, birth method, and breastfeeding), *C. difficile* colonization remained significantly higher only for infants in the cumulative exposure (AAHC) group (adjusted odds ratio (aOR): 1.50, 95% CI 1.03–2.17; *p* = 0.032), compared with those infants with no antibiotics and lower usage of cleaning products (NALC). *C. difficile* colonization was also higher in the AALC and NAHC infants, compared with the NALC infants, but did not attain statistical significance at *p* < 0.05 (Table 2).

#### 2.2.2. *C. difficile* Colonization at 12 Months of Age

In our study, the prevalence of *C. difficile* colonization in infants at 12 months of age was 46% (797/1728). *C. difficile* colonization rates in infants were different, depending on antimicrobial exposure: 41% for the NALC infants, 50% for the AALC infants, 44% for the NAHC infants, and 50% for the AAHC infants (*p* = 0.009; Figure 1). Before adjusting for covariates, the odds of colonization with *C. difficile* were 46% higher for both the AALC infants (OR: 1.46, 95% CI 1.12–1.88; *p* = 0.004) and the AAHC infants (OR: 1.46, 95% CI 1.12–1.90; *p* = 0.004), but not different for the NAHC infants (OR: 1.17, 95% CI 0.87–1.54; *p* = 0.256), compared with the NALC infants (Table 3). The covariates that were initially selected for testing in the models are shown in the directed acyclic graph (DAG; Appendix A). After adjusting for covariates in the final model (i.e., birth method, breastfeeding, and older siblingship), *C. difficile* colonization remained significantly higher in both the AALC infants (aOR: 1.36, 95% CI 1.02–1.83; *p* = 0.035) and the AAHC infants (aOR: 1.37, 95% CI 1.00–1.86; *p* = 0.043), compared with the NALC infants (Table 3). To account for antibiotic administration between 3 months and 12 months of age, we performed sensitivity analysis that adjusted models for oral antibiotic treatment during the 3 to 12 month period, as well as excluded infants with this usage. Similar results were obtained (Appendix A).

#### 2.2.3. Persistent *C. difficile* Colonization

In a smaller subset of 653 infants, 28% (184/653) were colonized with *C. difficile* at both 3–4 of age and 12 months of age. We classified this process as “persistent colonization”. Persistent *C. difficile* colonization was present in 18% of the NALC infants, 28% of the AALC infants, 26% of the NAHC infants, and 42% of the AAHC infants. Overall, antimicrobial exposure influenced persistent *C. difficile* colonization (*p* < 0.001). After adjusting for covariates in the final model (i.e., birth method, breastfeeding, and older siblingship), persistent *C. difficile* colonization remained significantly higher only in the AAHC infants (aOR: 2.40, 95% CI 1.33–4.35; *p* = 0.004), compared with the NALC infants (Table 4).

#### 2.2.4. Stratified Analysis of Antimicrobial Effects on *C. difficile* Colonization

In a subset of vaginally-delivered infants, *C. difficile* colonization was higher for the AAHC infants compared with the NALC infants at both 3–4 months of age and 12 months of age (aOR: 1.59, 95% CI 1.06–2.38; *p* = 0.025 and aOR: 1.62, 95% CI 1.15–2.29; *p* = 0.005, respectively). This association was not observed in caesarean-delivered infants, for whom there was already a greater risk of *C. difficile* colonization (Figure 2 and Figure 3; Appendix A). All caesarean-delivered infants were exposed to maternal IAP, so the reference group (NALC) for the vaginal-birth subanalysis consisted of infants who did not receive any direct antibiotics (Figure 2 and Figure 3). In a stratified analysis by sex, no difference was observed between boys and girls with respect to the impact of antimicrobial exposure on *C. difficile* colonization (Appendix A). In infants without an older sibling, *C. difficile* colonization at 12 months of age was higher for the AALC infants (aOR: 1.58, 95% CI 1.07–2.35; *p* = 0.021) and the AAHC infants (aOR: 1.77, 95% CI 1.16–2.70; *p* = 0.008), compared with the NALC infants. This association was not observed in infants who had an older sibling (Appendix A). Individual adjustments for each covariate showed that none were strong enough alone to remove the statistical significance from antimicrobial exposure on *C. difficile* colonization (Appendix A).

#### 2.2.5. Antimicrobial Exposure, *C. difficile* Colonization and Other Gut Microbes

Infant gut microbiota composition differed across groups of antimicrobial exposure at 3–4 months of age and to a lesser extent at 12 months of age. Of note, most changes occurred in the cumulative exposure group (AAHC), where the highest *C. difficile* colonization was observed. At 3–4 months of age, the relative abundance of *Bifidobacteriaceae* and *Bacteroidaceae* decreased, while the relative abundance of *Clostridaceae, Lachnospiraceae, Veillonellaceae*, and *Enterobacteriaceae* increased in the AALC and AAHC infants, compared with the NALC infants. In the NAHC infants, the relative abundance of *Lachnospiraceae* and *Ruminococcaceae* increased and the relative abundance of *Bifidobacteriaceae and Clostridaceae* decreased, compared with the NALC infants (Figure 4). At 12 months of age, the most obvious changes were a lower abundance of *Bacteroidaceae* in the AALC and AAHC infants, compared with the NALC infants, although *Clostridaceae, Lachnospiraceae, Veillonellaceae*, and *Enterobacteriaceae* were still higher. In the NAHC infants, the relative abundance of *Bifidobacteriaceae* and *Clostridaceae* decreased, compared with the NALC infants (Figure 4).

## 3. Discussion

### 3.1. Main Findings

In a general population of 1429 infants, *C. difficile* colonization of infant gut microbiota was affected by antimicrobial exposure. At 3–4 months of age, colonization rates were 1.5 times greater (95% CI 1.03–2.17; *p* = 0.032) following cumulative infant exposure to systemic and household antimicrobials (AAHC) than the colonization rates after minimal antimicrobial exposure (NALC). *C. difficile* colonization in infants aged 3–4 months persisted until the age of 12 months. Further, stratification by birth mode showed that in vaginally delivered infants, the colonization rates were 1.59 times greater (95% CI 1.06–2.38; *p* = 0.025) and 1.62 times greater (95% CI 1.15–2.29; *p* = 0.005) in the cumulative exposure group, compared with the rates in those with minimal antimicrobial exposure at 3–4 months of age and 12 months of age, respectively. This association was not observed in infants born via caesarean section, indicating that post-caesarean antimicrobial exposure did not further increase the risk of *C. difficile* colonization. At the older infant age, the association with antimicrobials was strongest from perinatal (maternal IAP and newborn IV) antibiotic exposure (odds ratio (OR):1.33, 95% confidence interval (CI) 1.10–1.62; *p* = 0.003). Although *C. difficile* colonization of infants is asymptomatic [15], its presence in gut microbiota at this early age is associated with future asthma and allergic diseases [20,21]. Our study is the first to evaluate both the systemic antibiotic exposure and the environmental antimicrobial exposure of infants, with separate consideration of infants delivered vaginally and by caesarean section.

### 3.2. Interpretation

Importantly, the cumulative effect of antimicrobial exposure on *C. difficile* colonization remained in vaginally-delivered infants, even after adjusting for covariates, showing that the vulnerability of full-term newborns to antibiotic exposure is not related to caesarean delivery. Consistent with what others have reported [15], we found that up to 31% of infants were colonized with *C. difficile* at a time when gut microbiota are being established. Antimicrobial exposure significantly increased *C. difficile* colonization at both 3–4 months of age and 12 months of age. The biological or clinical relevance of *C. difficile* colonization in infants is yet to be fully understood. In a small group of infants (*n* = 65), *C. difficile* was associated with an increased risk of allergic diseases in early childhood [20]. An earlier study of 957 infants linked the presence of *C. difficile* in infants at 1 month of age with atopic manifestations at 2 years of age [21]. Further, colonization with *C. difficile* in infancy may promote a dysbiotic gut environment by modifying the composition of the microbial ecosystem [18], or serve as a reservoir for adult *C. difficile* infection [16,17]. Early microbial dysbiosis may also be associated with inflammatory bowel disease (IBD), allergy and asthma, obesity, and other metabolic disorders [19].

Several studies have evaluated the impact of maternal IAP or postnatal antibiotics on infant gut microbiota [7,8,22]. Similar to our results, Tapiainen et al. [8] reported changes in infant gut microbiota from both IAP exposure and IV antibiotics that were still observed at 6 months of age, including the enrichment of *Clostridium* and the depletion of *Bacteroides* species. Consistent with what others have reported [8,23,24,25,26], we found a reduction of *Bifidobacteriaceae* and *Bacteroidaceae* in gut microbiota following antibiotic exposure, as well as an increase in *Clostridaceae, Lachnospiraceae, Veillonellaceae*, and *Enterobacteriaceae* at both 3–4 months of age and 12 months of age. Colonization of more *Proteobacteria* (phylum to which *Enterobacteriaceae* belongs) may be a signal for gut dysbiosis and inflammation [27], while a reduction in important gut microbes may provide room for *C. difficile* colonization and overgrowth. We observed the most significant changes in infants with the highest antimicrobial exposure (the AAHC group), who were also identified as having the highest *C. difficile* colonization.

While many household cleaning products contain antimicrobials that can give rise to resistant bacteria [28], the evidence of their impact on infant gut microbiota is limited. Consistent with a previous report on frequent use of household disinfectants in a smaller sample of CHILD study infants [9], this study found a higher abundance of *Lachnospiraceae* in infants of 3–4 months of age who were exposed to the higher usage of cleaning products. Moreover, our results further demonstrated that the combined effect of systemic antibiotic exposure and additional antimicrobial exposure from household cleaning products increases the likelihood of *C. difficile* colonization. This cumulative antimicrobial effect on *C. difficile* colonization at 3–4 months of age persisted until at least 12 months of age, demonstrating that the collateral damage inflicted on the gut microbiota is not rapidly repaired [29]. This effect was even greater in infants who were persistently colonized with *C. difficile* at both 3–4 months of age and 12 months of age.

We found few other early life variables that influenced *C. difficile* colonization at 3–4 months of age or 12 months of age. As previously reported by others [30,31], infants born via emergency caesarean section (CS) were more likely to be colonized by *C. difficile* at 3–4 months of age, compared with vaginally delivered infants. Many women who undergo emergency CS delivery planned to give birth vaginally and are positive for GBS [32], increasing the cumulative antimicrobial exposure of the newborn. Previous epidemiological studies have identified breastfeeding as an important contributor to the infant gut microbiota [22], [31,33]. In our study, exclusively breastfed infants were less likely to be colonized with *C. difficile* at 3–4 months of age than were mixed-fed infants or exclusively formula-fed infants. We found that having an older sibling seemed to strongly influence *C. difficile* colonization at 12 months of age. Infants with an older sibling had lower *C. difficile* colonization. This finding was similar to that of a systematic review comprising six studies that reported a decreased abundance of *Clostridium* in infants with an older sibling [34]. It is suggested that the influence of other children or siblings on the infant gut microbiota is in line with the “hygiene hypothesis”, in that other children or siblings increase the infant’s exposure to early gut colonizers that prime the infant’s immune system and provide colonization resistance against pathogens and *C. difficile* [33,35].

### 3.3. Strengths and Limitations

Our study has several strengths, including the application of high throughput sequencing and qPCR to profile gut microbiota and *C. difficile* colonization in a prospective birth cohort with representative and large sample size. To the best of our knowledge, this study is the first to examine the influence of household cleaning products and their cumulative effect, together with systemic antibiotics, on *C. difficile* colonization and the infant gut microbiota. Unique to our study was the capture of both perinatal (maternal IAP and newborn IV) and postnatal antibiotic use. We also performed a sensitivity analysis for *C. difficile* colonization at age 12 months to account for antibiotics given during the period from 3 months to 12 months of age.

Our findings should be considered with some limitations. Exposure status to cleaning products was dependent on self-reported questionnaires, and we were not able to determine the specifics of cleaning product brands, ingredients, or how much product was used per application. In addition, we were not able to perform analysis based on chemical composition. However, we assigned a score based on the frequency of use of each product and classified all infants as either living in a home with higher (above average) or lower (below average) usage of cleaning products. Moreover, we did not report on the longitudinal colonization of *C. difficile*, as our sample collection was limited to two time points. However, we performed analysis to examine persistent colonization at both 3–4 months of age and 12 months of age.

## 4. Materials and Methods

### 4.1. Study Design

This study included a subsample of 1429 infants at 3–4 months of age and 1728 infants at 12 months of age from families that were enrolled in the Canadian Healthy Infant Longitudinal Development (CHILD) Cohort (www.childstudy.ca; accessed on 10 January 2022). Women were enrolled into CHILD during the second or third trimester of pregnancy between 2009 to 2012, from study sites in Vancouver, Edmonton, and Manitoba. Written informed consent was obtained from the mothers upon enrollment. This study was approved by the Human Research Ethics Boards of the University of Alberta, the University of Manitoba, and the University of British Columbia.

### 4.2. Exposures

This prospective population-based cohort study examined the effect of antimicrobial exposures, before or at 3 months of age, on *C. difficile* colonization in the gut of the infants. Data on maternal intrapartum antibiotic prophylaxis (IAP) and newborn antibiotic treatment were obtained from hospital birth charts for each of the participants. At 3 months post-partum, the mothers completed validated questionnaires regarding infant usage of antibiotics and the frequency of use of various household cleaning products. Parents were asked about their usage of household cleaning products from a list of 26 cleaning products (Appendix A). The cleaning products questionnaire was validated with visual inspection of the products by research assistants during home visits [9]. The frequency of use for each product was assigned a score: 0 for never (not used), 1 for less than a month usage, 2 for monthly usage, 3 for weekly usage, and 4 for daily usage. The scores for each respondent were added together to obtain a total score. The total scores were split at the median into two groups of higher usage (i.e., living in a home with above average cleaning product use) and lower usage (i.e., living in a home with below average cleaning product use) of household cleaning products to make comparisons of the effect of sizes during the analyses. To determine the impact of the antibiotics and the household cleaning products, infants were assigned to one of four groups: (1) no antibiotic exposure and lower usage of cleaning products (**NALC**), (2) any antibiotic exposure and lower usage of cleaning products (**AALC**), (3) no antibiotic exposure and higher usage of cleaning products (**NAHC**) and (4) any antibiotic exposure and higher usage of cleaning products (**AAHC**). The exposure to “any” antibiotics indicated exposure to maternal IAP and/or the administration of antibiotics directly to infants from birth to 3 months of age.

### 4.3. Fecal Microbiota Analysis

Infant stool samples (fresh or frozen) were collected at approximately 3–4 months of age and 12 months of age after a home assessment or in the clinic. Samples were stored at −80 °C prior to analysis. Fecal samples were characterized with Illumina MiSeq using the bacterial 16S ribosomal ribonucleic acid (rRNA) gene hypervariable V4 region, as previously described [36]. Analysis of *C. difficile* was performed using quantitative polymerase chain reaction (qPCR) with appropriate primers, as previously described [7]. Primers and probe efficiency were determined by a standard curve procedure by establishing five 1:10 serial dilutions of *C. difficile* American Type Culture Collection (ATCC) 9689D-5 genomic DNA, starting at 1 ng/uL. For each plate, a non-template control was used. An efficiency between 90% and 110%, and an R^2^ greater than or equal to 0.9 for the primers and probes combination were used as quality control parameters for each run. A Quantitative Insights Into Microbial Ecology (QIIME) pipeline (www.qiime.org, accessed on 15 June 2017) was used to group microbiota into taxonomic order and to summarize Operational Taxonomic Unit (OTU) data within infant fecal samples.

### 4.4. Statistical Analysis

*C. difficile* colonization (outcome) at 3–4 months of age and 12 months of age was analyzed as a binary variable (yes or no). Non-parametric tests were carried out as appropriate to compare *C. difficile* colonization, taxon mean relative abundance, antimicrobial exposure, and demographic variables. Statistical significance was defined as a two-sided *p* or *q*-value ≤ 0.05, after a false discovery rate (FDR) correction for multiple comparisons. Potential confounding variables were identified from the literature [31,33,37]. They included maternal age and race, family income, birth method, gestational age, breastfeeding status at 3 months of age, furry pet ownership, having an older sibling(s), and tobacco smoke exposure from birth to 3 months of age. Thereafter, a reduced set of these variables was selected using the directed acyclic graph (DAG) method; these variables were tested in models to prevent over-adjustment [38] (See Appendix A). Logistic regression analysis was used to determine the association between antimicrobial exposure and *C. difficile* colonization. Model 1 was adjusted for maternal and birth characteristics (maternal age and birth method), model 2 was adjusted for postnatal characteristics (breastfeeding, furry pet, older siblingship, and smoke exposure). Model 3 was adjusted for variables from models 1 or 2 that had a *p*-value ≤ 0.05 or caused a ≥15% change in the estimate of antimicrobial exposure. Confounding variables from model 3 were retained in the final model if they had a *p*-value ≤ 0.05 or if they caused a ≥15% change in the estimate of antimicrobial exposure. It is noteworthy that model 3 was not included for the 12-month analysis, because most covariates were not significant at *p*-value ≤ 0.05 in models 1 or 2; confounding variables retained in the final model were those from models 1 or 2 that had a *p*-value ≤ 0.05 or caused a ≥15% change in estimate of antimicrobial exposure. Statistical analysis was conducted using STATA 13.0 software (64-bit); Stata Corp 4905 Lakeway Drive College Station, Texas 77845 USA.

## 5. Conclusions

In Canada, oral antibiotic treatment of young infants is not common; however, indirect neonatal exposure to intrapartum antibiotics administered to the mother is increasing, due to current recommendations for GBS and the growing prevalence of caesarean delivery. Our results support ongoing efforts to reduce antimicrobial exposure during infancy, especially in the neonatal period. These results show that cumulative exposure to antibiotics and household cleaning products (environmental antimicrobials) is not without consequence. Previous epidemiological studies have linked early life antimicrobial exposure to the development of childhood asthma and allergic diseases, but the mechanisms for these associations are unknown. *C. difficile* colonization and/or gut microbiota composition may or may not be a pathway. Further studies are required to replicate these findings in other populations and to determine the impact on future health outcomes.

## Figures and Tables

**Figure 1 antibiotics-11-00981-f001:**
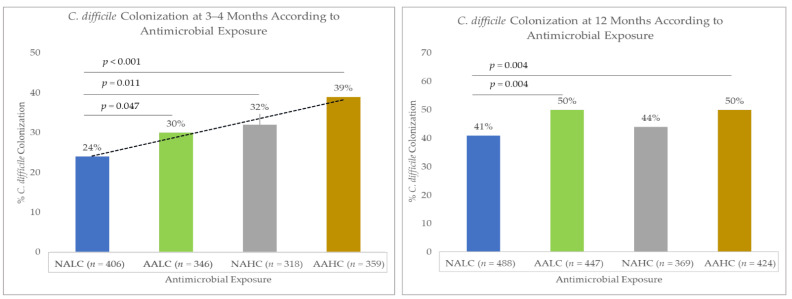
*C. difficile* colonization according to antimicrobial exposure (NALC: no antibiotics and lower usage of cleaning products; AALC: any antibiotics and lower usage of cleaning products; NAHC: no antibiotics and higher usage of cleaning products; AAHC: any antibiotics and higher usage of cleaning products).

**Figure 2 antibiotics-11-00981-f002:**
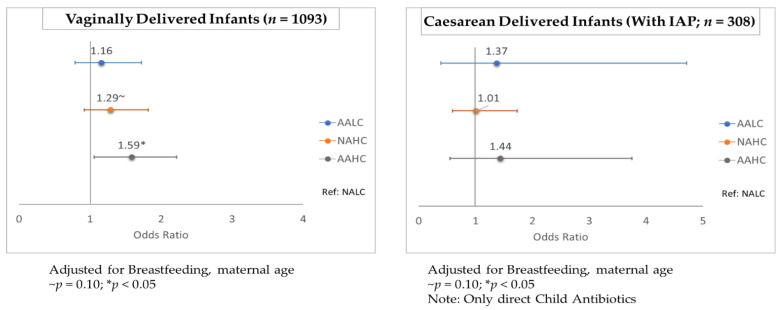
Stratified analysis by birth method for antimicrobial exposure and *C. difficile* colonization at 3–4 months of age. (NALC: no antibiotics and lower usage of cleaning products; AALC: any antibiotics and lower usage of cleaning products; NAHC: no antibiotics and higher usage of cleaning products; AAHC: any antibiotics and higher usage of cleaning products).

**Figure 3 antibiotics-11-00981-f003:**
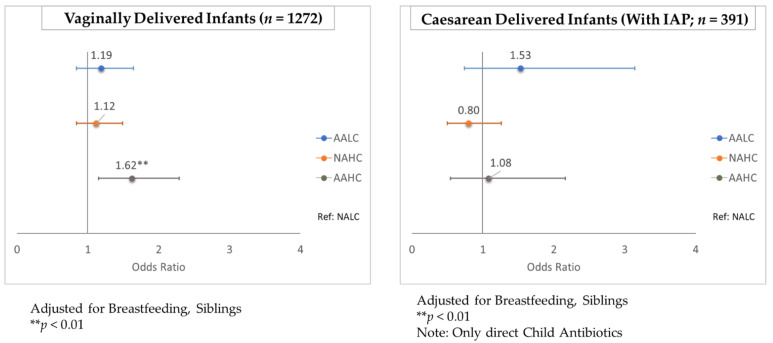
Stratified analysis by birth method for antimicrobial exposure and *C. difficile* colonization at 12 months of age. (NALC: no antibiotics and lower usage of cleaning products; AALC: any antibiotics and lower usage of cleaning products; NAHC: no antibiotics and higher usage of cleaning products; AAHC: any antibiotics and higher usage of cleaning products).

**Figure 4 antibiotics-11-00981-f004:**
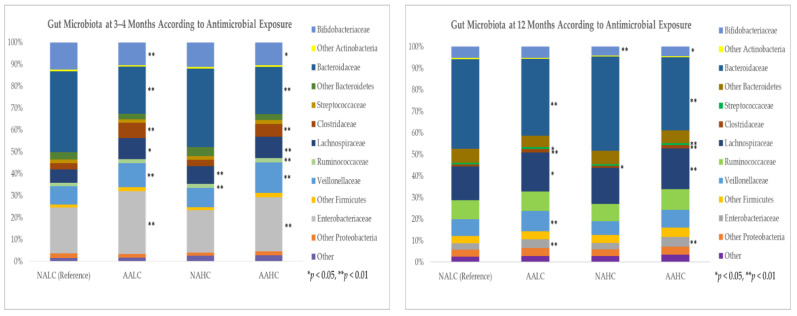
Stacked bar plots of mean relative abundance of dominant taxa (family level) according to antimicrobial exposure. Comparisons were performed using Kruskal–Wallis test with Dunn’s post hoc test. Positive false discovery rate (FDR) was used to adjust *p*-values for multiple testing.

**Table 1 antibiotics-11-00981-t001:** Population characteristics according to antimicrobial exposure at 3 months (3–4 months-old sample; N = 1429).

Row Percentages	Total ^a^	NALC * n = 406	AALC * n = 346	NAHC * n = 318	AAHC * n = 359	*p* Value
		29%	24%	22%	25%	
**Maternal age**						
18–29	503	148 (29%)	105 (21%)	137 (27%)	113 (22%)	**<0.001**
30–39	874	241 (26%)	220 (25%)	177 (20%)	236 (27%)	
≥40	52	17 (33%)	21 (40%)	4 (8%)	10 (19%)	
**Maternal race**						
Caucasian	1072	270 (25%)	271 (25%)	246 (23%)	285 (27%)	**<0.001**
Asian	203	108 (53%)	38 (18%)	33 (16%)	24 (12%)	
Other	141	27 (19%)	34 (24%)	35 (25%)	45 (32%)	
**Family income**						
<50,000	199	34 (17%)	31 (16%)	54 (27%)	80 (40%)	**<0.001**
50,000–99,999	487	134 (27%)	121 (25%)	111 (23%)	121 (25%)	
≥100,000	516	207 (40%)	158 (31%)	105 (20%)	46 (9%)	
Prefer not to answer	130	26 (20%)	25 (19%)	33 (26%)	46 (35%)	
**Birth method**						
Vaginal	1096	406 (37%)	211 (19%)	317 (29%)	162 (15%)	**<0.001**
CS-elective	134	0	52 (39%)	0	82 (61%)	
CS-emergency	194	0	82 (42%)	0	112 (58%)	
**Gestational age**						
<39 weeks	372	73 (20%)	107 (29%)	73 (20%)	119 (32%)	**<0.001**
≥39 weeks	1052	333 (32%)	238 (23%)	245 (23%)	236 (22%)	
**Infant Sex**						
Male	766	206 (27%)	196 (26%)	161 (21%)	203 (27%)	0.172
Female	663	200 (30%)	150 (23%)	157 (24%)	156 (24%)	
**Breastfeeding at 3 Months**						
Exclusive	791	262 (33%)	209 (26%)	153 (19%)	167 (21%)	**<0.001**
Mixed	384	98 (26%)	88 (23%)	85 (22%)	113 (29%)	
Formula	251	45 (18%)	48 (19%)	80 (32%)	78 (31%)	
**Furry pet**						
No	777	258 (33%)	205 (26%)	258 (20%)	205 (21%)	**<0.001**
Yes	648	146 (23%)	141 (22%)	166 (26%)	195 (30%)	
**Older sibling**						
No	712	189 (27%)	194 (27%)	125 (18%)	204 (29%)	**<0.001**
Yes	712	216 (30%)	150 (21%)	193 (27%)	153 (21%)	
**Smoke exposure**						
No	1160	352 (30%)	290 (25%)	251 (22%)	267 (23%)	**<0.001**
Yes	248	51 (21%)	50 (20%)	62 (25%)	85 (34%)	

* Antimicrobial exposure (NALC: no antibiotics and lower usage of cleaning products; AALC: any antibiotics and lower usage of cleaning products; NAHC: no antibiotics and higher usage of cleaning products; AAHC: any antibiotics and higher usage of cleaning products). **^a^** Total may not add up due to missing data; IAP: intrapartum antibiotic prophylaxis; *p*-value calculated using Pearson chi-square test and displayed in **bold** when statistically significant.

**Table 2 antibiotics-11-00981-t002:** Univariable and multivariable logistic regression for antimicrobial exposure and *C. difficile* colonization at 3–4 months of age.

	Crude (Unadjusted)		Model 1 (Adjusted for Maternal and Birth Characteristics)		Model 2 (Adjusted for Postnatal Characteristics)		Model 3 (Adjusted for Maternal Age, Birth Method, Breastfeeding, Furry Pet, Older Siblingship, and Smoke Exposure)		Final Model (Adjusted for Maternal Age, Birth Method, and Breastfeeding)	
	OR (95% CI)	*p* Value	aOR (95% CI)	*p* Value	aOR (95% CI)	*p* Value	aOR (95% CI)	*p* value	aOR (95% CI)	*p* Value
**Antimicrobial Exposure * (ref = NALC)**										
AALC	1.38 (1.00–1.91)	**0.047**	1.23 (0.86–1.76)	0.238	1.29 (0.92–1.80)	0.136	1.17 (0.81–1.69)	0.376	1.21 (0.84–1.73)	0.293
NAHC	1.52 (1.10–2.11)	**0.011**	1.48 (1.06–2.06)	**0.020**	1.22 (0.86–1.72)	0.252	1.19 (0.84–1.69)	0.304	1.27 (0.90–1.78)	0.166
AAHC	2.03 (1.49–2.78)	**<0.001**	1.67 (1.16–2.41)	**0.006**	1.53 (1.10–2.14)	**0.010**	1.33 (0.90–1.95)	0.141	1.50 (1.03–2.17)	**0.032**
**Block 1: Maternal and birth characteristics**										
**Maternal age (ref = 30–39)**										
18–29	1.60 (1.34–2.13)	**<0.001**	1.79 (1.41–2.27)	**<0.001**			1.66 (1.29–2.14)	**<0.001**	1.71 (1.34–2.18)	**<0.001**
≥40	0.80 (0.41–1.55)	0.513	0.79 (0.40–1.55)	0.505			0.90 (0.45–1.79)	0.768	0.86 (0.43–1.71)	0.688
**Birth method (ref = Vaginal)**										
CS-elective	1.42 (1.02–2.15)	**0.039**	1.44 (0.93–2.22)	0.095			1.29 (0.82–2.03)	0.260	1.28 (0.83–1.99)	0.251
CS-emergency	1.63 (1.19–2.24)	**0.002**	1.54 (1.06–2.24)	**0.022**			1.50 (1.02–2.22)	**0.038**	1.49 (1.02–2.18)	**0.038**
**Gestational age (ref ≤ 39weeks)**										
**≥39 weeks**	1.00 (0.77–1.29)	0.979	1.08 (0.83–1.41)	0.550						

**Block 2: Postnatal characteristics**										
**Breastfeeding ^a^ (ref = Exclusive)**										
Mixed	1.96 (1.50–2.55)	**<0.001**			1.88 (1.43–2.47)	**<0.001**	1.89 (1.43–2.49)	**<0.001**	1.88 (1.43–2.46)	**<0.001**
Formula	3.11 (2.31–4.19)	**<0.001**			2.78 (2.03–3.80)	**<0.001**	2.65 (1.93–3.63)	**<0.001**	2.71 (1.99–3.69)	**<0.001**
**Furry pet (ref=No)**										
Yes	1.38 (1.10–1.73)	**0.005**			1.17 (0.92–1.50)	0.180	1.19 (0.93–1.52)	0.148		
**Older sibling (ref = No)**										
Yes	0.77 (0.62–0.97)	**0.030**			0.80 (0.63–1.02)	0.073	0.87 (0.68–1.13)	0.316		
**Smoke exposure (ref = No)**										
Yes	1.77 (1.33–2.35)	**<0.001**			1.43 (1.06–1.93)	**0.018**	1.31 (0.97–1.78)	0.075		

* Antimicrobial exposure in infants by 3 months of age (NALC: no antibiotics and lower usage of cleaning products; AALC: any antibiotics and lower usage of cleaning products, NAHC: no antibiotics and higher usage of cleaning products; AAHC: any antibiotics and higher usage of cleaning products). ^a^ Breastfeeding at 3 months of age; OR: odds ratio, aOR: adjusted odds ratio; CI: confidence interval; statistically significant *p*-values displayed in **bold**.

**Table 3 antibiotics-11-00981-t003:** Univariable and multivariable logistic regression for antimicrobial exposure and *C. difficile* colonization at 12 months of age.

	Crude (Unadjusted)		Model 1 (Adjusted for Birth Characteristics)		Model 2 (Adjusted for Postnatal Characteristics)		Final Model (Adjusted for Birth Method, Breastfeeding, and Older Siblingship)	
	OR (95% CI)	*p* Value	aOR (95% CI)	*p* Value	aOR (95% CI)	*p* Value	aOR (95% CI)	*p* Value
**Antimicrobial Exposure *** **(ref = NALC)**								
AALC	1.46 (1.12–1.88)	**0.004**	1.44 (1.08–1.92)	**0.012**	1.34 (1.02–1.75)	**0.030**	1.36 (1.02–1.83)	**0.035**
NAHC	1.17 (0.89–1.54)	0.256	1.16 (0.88–1.53)	0.272	1.13 (0.85–1.51)	0.371	1.11 (0.84–1.48)	0.442
AAHC	1.46 (1.12–1.90)	**0.004**	1.47 (1.09–1.99)	**0.012**	1.37 (1.04–1.80)	**0.023**	1.37 (1.00–1.86)	**0.043**
**Block 1:** **Birth characteristics**								
**Birth method (ref = Vaginal)**								
CS-elective	1.17 (0.86–1.61)	0.305	0.90 (0.63–1.28)	0.575			1.13 (0.78–1.63)	0.511
CS-emergency	1.22 (0.93–1.59)	0.148	0.97 (1.71–1.33)	0.892			0.86 (0.63–1.18)	0.374
**Gestational age (ref ≤ 39weeks)**								
≥39 weeks	0.81 (0.66–1.01)	0.067	0.85 (0.68–1.06)	0.170				

**Block 2: Postnatal characteristics**								
**Breastfeeding ^a^ (ref = Yes)**								
No	1.10 (0.91–1.34)	0.297			1.09 (0.89–1.33)	0.369	1.08 (0.89–1.32)	0.399
**Furry pet** **(ref = No)**								
Yes	0.89 (0.73–1.07)	0.236			0.84 (0.68–1.02)	0.091		
**Older sibling (ref = No)**								
Yes	0.60 (0.49–0.72)	**<0.001**			0.60 (0.49–0.73)	**<0.001**	0.59 (0.48–0.73)	**<0.001**
**Smoke Exposure (ref = No)**								
Yes	0.94 (0.73–1.22)	0.690			0.91 (0.70–1.19)	0.533		

Notes: * Antimicrobial exposure in infants by 3 months of age (NALC: no antibiotics and lower usage of cleaning products; AALC: any antibiotics and lower usage of cleaning products; NAHC: no antibiotics and higher usage of cleaning products; AAHC: any antibiotics and higher usage of cleaning products); ^a^ Breastfeeding at 12 months; OR: odds ratio, aOR: adjusted odds ratio; CI: confidence interval; statistically significant *p*-values displayed in **bold**.

**Table 4 antibiotics-11-00981-t004:** Univariable and multivariable logistic regression for antimicrobial exposure and persistent *C. difficile* colonization.

	Crude (Unadjusted)		Model 1 (Adjusted for Maternal and Birth Characteristics)		Model 2 (Adjusted for Postnatal Characteristics)		Model 3 (Adjusted for Maternal Age, Birth Method, Breastfeeding, Older Siblingship, and Smoke Exposure)		Final Model (Adjusted for Birth Method, Breastfeeding, and Older Siblingship)	
	OR (95% CI)	*p* Value	aOR (95% CI)	*p* Value	aOR (95% CI)	*p* Value	aOR (95% CI)	*p* Value	aOR (95% CI)	*p* Value
**Antimicrobial Exposure * (ref = NALC)**										
AALC	1.75 (1.06–2.89)	**0.028**	1.57 (0.91–2.71)	0.104	1.38 (0.81–2.35)	0.235	1.33 (0.75–2.37)	0.322	1.31 (0.74–2.13)	0.337
NAHC	1.59 (0.95–2.67)	0.075	1.61 (0.96–2.71)	0.071	1.36 (0.77–2.39)	0.481	1.39 (0.79–2.44)	0.241	1.50 (0.87–2.57)	0.142
AAHC	3.20 (1.96–5.23)	**<0.001**	2.77 (1.55–4.92)	**0.001**	2.55 (1.51–4.33)	**<0.001**	2.42 (1.32–4.44)	**0.004**	2.40 (1.33–4.35)	**0.004**
**Block 1: Maternal and birth characteristics**										
**Maternal age (ref = 30–39)**										
18–29	1.48 (1.04–2.13)	**<0.030**	1.61 (1.11–2.34)	**0.011**			1.24 (0.82–1.85)	0.294		
≥40	0.76 (0.27–2.09)	0.598	0.74 (0.26–2.11)	0.584			0.99 (0.34–2.92)	0.998		
**Birth method (ref = Vaginal)**										
CS-elective	1.66 (0.94–2.94)	0.079	1.03 (0.53–2.01)	0.921			1.21 (0.60–2.45)	0.579	1.19 (0.59–2.39)	0.609
CS-emergency	2.10 (1.32–3.35)	**0.002**	1.51 (0.87–2.61)	0.141			1.18 (0.65–2.12)	0.574	1.16 (0.65–2.06)	0.601
**Gestational age (ref ≤ 39 weeks)**										
≥39 weeks	0.75 (0.51–1.10)	0.151	0.77 (0.52–1.15)	0.217						

**Block 2: Postnatal characteristics**										
**Breastfeeding ^a^ (ref = Exclusive)**										
Mixed	2.30 (1.53–3.44)	**<0.001**			2.42 (1.58–3.72)	**<0.001**	2.35 (1.53–3.62)	**<0.001**	2.22 (1.45–3.39)	**<0.001**
Formula	3.34 (2.11–5.28)	**<0.001**			2.55 (1.55–4.21)	**<0.001**	2.44 (1.48–4.03)	**<0.001**	2.83 (1.74–4.58)	**<0.001**
**Furry pet (ref = No)**										
Yes	1.26 (0.89–1.77)	0.180			0.93 (0.63–1.37)	0.749				
**Older sibling (ref = No)**										
Yes	0.38 (0.26–0.54)	**<0.001**			0.36 (0.24–0.54)	**<0.001**	0.38 (0.25–0.58)	**<0.001**	0.40 (0.27–0.59)	**<0.001**
**Smoke exposure (ref = No)**										
Yes	1.63 (1.05–2.54)	**0.029**			1.58 (0.97–2.59)	0.064	1.50 (0.92–2.45)	0.103		

* Antimicrobial exposure in infants by 3 months of age. (NALC: no antibiotics and lower usage of cleaning products; AALC: any antibiotics and lower usage of cleaning products; NAHC: no antibiotics and higher usage of cleaning products; AAHC: any antibiotics and higher usage of cleaning products); ^a^ Breastfeeding at 3 months of age; OR: odds ratio, aOR: adjusted odds ratio; CI: confidence interval; statistically significant *p*-values displayed in **bold**.

## Data Availability

The data and the analysis code that support the findings of this study can be made available from the corresponding author and from CHILD Cohort Study coordinators upon reasonable request. These data, including study participant data, are securely stored in the https://childdb.ca database; accessed on 10 January 2022.

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
