# Peer review of "Early Life Antimicrobial Exposure: Impact on Clostridioides difficile Colonization in Infants"

_antibiotics, 2022, doi:10.3390/antibiotics11070981_

Round 1
Reviewer 1 Report
This article by Obiakor et al. utilizes the extensive CHILD cohort database to present an interesting and well-written analysis of antibiotic use and household cleaner exposure in relation to infant gut colonization by C. difficile. The authors report increased colonization of C. difficile with infant antibiotic exposure at both 3-4 months and 12 months of age and several differences in microbiota relative abundances; differences associated with antibiotic use were exacerbated by high exposure to household cleaning agents. This analysis reveals important relationships between antibiotics and the potential for higher risk of C. difficile-related illness and future studies can confirm these changes in gut microbiota colonization are related to later-life health outcomes. This reviewer only has minor comments.
1. The variables included in each of the models listed in Tables 2, 3, and 4 are not entirely clear to me. Within each table, it seems that block 1 corresponds to model 1 and same for block 2 and model 2, with model 3 incorporating the variables from both block 1 and block 2. However, what variables are included in the final model? Does the final model include the significant variables for models 1, 2, or 3? Or are the variables in the final model listed in their respective DAGs? It would be helpful if the variables for each model were explicitly stated in the methods.
2. Why does Table 3 (12 month timepoint analysis) not include a model 3 in the analysis like Tables 2 and 4? It seems like a "model 3" could be built for Table 3, so I am curious as to why it was not included in the analysis.
Author Response
"Please see the attachment."

Reviewer 2 Report
Please see attached comments.

Author Response
"Please see the attachment"
